# Learning Clinical Skills Using Haptic vs. Phantom Head Dental Chair Simulators in Removal of Artificial Caries: Cluster-Randomized Trials with Two Cohorts’ Cavity Preparation

**DOI:** 10.3390/dj10110198

**Published:** 2022-10-24

**Authors:** Jonathan P. San Diego, Tim J. Newton, Anika K. Sagoo, Tracy-Ann Aston, Avijit Banerjee, Barry F. A. Quinn, Margaret J. Cox

**Affiliations:** 1Faculty of Dentistry, Oral & Craniofacial Sciences, King’s College London, London SE1 1UL, UK; 2Institute of Life Course and Medical Sciences, University of Liverpool, Liverpool L7 8TX, UK

**Keywords:** haptics, simulation, dental-task trainer, virtual reality, dental education

## Abstract

Dental task trainer simulators using haptics (virtual touch) offers a cost-effective method of teaching certain clinical skills. The purpose of this study is to evaluate students’ performance in removing artificial caries after training with either a haptic dental chair simulator with virtual reality or a traditional dental chair simulator with a mannequin head. Cluster Randomized Controlled Trials in two cohorts, both Year 1 dental students. Students taught using traditional dental chair simulators were compared with students taught using haptic-based simulators on their ability to cut a cavity in a plastic tooth following training. Across both cohorts, there was no difference in the quality of cavity cut, though students’ technique differed across the two simulator groups in some respects. No difference was seen across both cohorts in the quality of cavity cut for a simple preparation, though students in the haptic condition performed less well in the more demanding task. Moreover, students in the haptic group were also less likely to be perceived to be ‘holding the instrument appropriately’. These findings suggest further investigation is needed into the differences in handling of instruments and level of clinical task difficulty between the simulators.

## 1. Introduction

Simulation-based learning systems provide a safe and practical approach for pre-clinical students to practise and gain the basic operative and core clinical concepts mandated by the profession’s governing body, the General Dental Council (GDC), without compromising patient safety. Seropian and colleagues [1] categorized these simulation systems into four separate groups: plastic-based nondynamic simulators; plastic-based dynamic simulators; virtual reality simulators with low-fidelity haptics; and virtual reality simulators with high-fidelity haptics [2].

Mannequins with computers have most commonly been implemented in dental, nursing and medical education and can be manipulated to imitate various clinical and surgical scenarios and allow repeated clinical practice in a safe setting [3,4]. Moreover, simulation-based training has also been perceived as a valuable tool in allied health professional (AHP) education. From the limited number of studies conducted, simulation-based learning activities have been shown to have a positive impact on short-term skill enhancement and increases AHP students’ confidence in their knowledge and skill level following use. However, further research is needed to understand the long-term effectiveness of simulation-based training in preparing AHP students for clinical practice [5,6,7,8,9,10,11,12]. Before becoming widely prominent in clinical education, simulation technology was most notably capitalized in other high-stakes environments, such as aviation and aerospace training [13,14]. In the late 1800s, dental schools incorporated bench-top simulators alongside resin-based teeth into their clinical training sessions. However, with the lack of realism in those early days, the learning environment provided was inadequate. In 1894, the first historic phantom head simulator was developed and designed by Oswald Fergus with the purpose of providing dental students with a training model that resembled a real patient scenario. This would subsequently help to facilitate their acquisition of essential restorative dentistry skills and resultantly prepare students for professional practice [15]. The phantom head simulator has extensively evolved from what started as a low-fidelity model to systems with high-fidelity status and to this day is the main effective training tool in pre-clinical dental education. For the last decade, simulation-based mannequins have been described as near to being ‘ergonomically accurate’ and have proven to establish a learning environment that enhances the teaching, learning and practise of certain motor skills and precision [15]. However, mannequin-based learning systems require constant supervision, which can impact faculty teaching, the assessment of student performance and managing the safe use of the simulators; for example, navigating ‘electrically driven hand-pieces’ of phantom head simulators. This, paired with continuous changes in the curriculum, can introduce time and support constraints affecting the ability to effectively teach clinical skills to students to ensure they are up to qualifying standard [15,16,17]. Moreover, the implementation of many new complex operative techniques into the pre-clinical training cannot be demonstrated with traditional clinical simulator systems. In effect, they do not allow the practice and judgement of some characteristics/skills which can be seen in complex patient cases or are crucial to accurately performing procedures, for example, hand-eye coordination [17]. With the requirement to be competent in increasingly complex procedures, standalone phantom head simulators do not offer an effective solution and technological advances are required to address their limitations.

This is where simulators that are coupled with virtual reality (VR) and haptic technologies come in, integrating an immersive virtual 3D experience and touch feedback into the simulation [18,19]. Haptic technology creates the perception of touch by administering mechanical pressure/forces to the users on an interface, stimulating the tactile receptors in the skin which in turn triggers motor function. Learning how to utilize tactile skills in dental procedures is a crucial component of dental education. As dental surgical techniques become more sophisticated, there is an increasing demand from patients to have procedures which are ‘less radical and intrusive’ and requires clinicians to be more dependent on sensory perception as opposed to direct visual perception of patients [15]. Haptic technology is claimed to have greater accuracy when assessing student clinical skills compared to traditional training methods by being able to measure characteristics that the traditional methods cannot (such as efficiency of movement), which in turn provides greater student satisfaction [20,21,22].

It has been suggested that introducing psychomotor skills early in training can enhance dental students’ ability to become competent clinicians [23]. Due to the great success of stand-alone phantom head simulators, it would be reasonable to suggest that as a benchmark, student scores achieved following training with VR-haptic simulators should resonate with their scores obtained from phantom head simulation training. Currently there are very few studies available; however, of those, it is suggested that VR-haptic simulators are of equal value to measuring clinical ability in pre-clinical skills’ training to traditional training simulators [24,25]. For example, Dwisaptarini and colleagues had shown that identical scores were achieved during the examination of the effectiveness of a visuo-tactile VR task simulator for training sixth year dental students in a minimally invasive caries removal exercise compared to extracted teeth [26]. Moreover, a recent study shows that initial training on either a haptic VR trainer or on conventional plastic analogue teeth for cavity preparation training had similar impact on first year dental students learning curves [27]. Additionally, data retrieved by AI-Saud and Colleagues [28] indicated that haptic task trainers utilized in the early stages of undergraduate dental study have a greater potential of predicting subsequent clinical performance scores compared to a traditional phantom head.

Additionally, other factors, such as the type of feedback that can be provided by VR-haptic simulators, can further enhance the learning process of skill acquisition. Suebnukarn and colleagues [29] found that students who trained and completed a crown preparation and endodontic access opening task on a VR haptic simulator obtained higher scores if provided with ‘Knowledge of Performance’ (KP) feedback in addition to ‘Knowledge of Result’ (KR) feedback. KP analyses the users’ augmented kinematic parameters, whereas KR is information based on only the simulated clinical outcome. This indicates that feedback which focuses on the user’s patterns of movement strengthens early skill acquisition and retention compared to traditional simulation outcome feedback.

An experimental study by Nilsson et al., 2007, evaluated and compared the effectiveness of VR simulation training and the traditional training method on improving students’ abilities to analyse and accurately interpret visual-spatial information on radiographs. Results showed that showed that students with low visual-spatial abilities benefited more from VR-simulation training compared to traditional method training [30]. These results suggest that one’s visual spatial awareness may be a factor that influences the effectiveness of VR simulation training.

An interdisciplinary team of educationalists, cyberneticists, social scientists and engineers from three universities developed an award-winning haptic dental task training system that utilizes VR and haptic technology. By wearing 3D glasses and operating a haptic device (that navigates a virtual dental drill), users can learn and practise aspects of cavity preparations at different levels of difficulty in a 3D virtual environment, where they will experience audio tactile sensations, such as ‘feeling the different layers of a tooth when drilling and hearing the drill’ [31].

This paper aims to evaluate the effectiveness of the haptic dental task trainer simulator compared to a traditional mannequin simulator on learning some aspects of cavity preparation skills by comparing students’ performances. In this study, two groups were trained either traditionally or haptically and then tested using the traditional phantom head simulator. We hypothesize that there would be no difference in the performance between the two simulator conditions in removing artificial caries based on specific marking criteria.

## 2. Materials and Methods

The study has been informed by the evaluation of the traditional teaching and training practices and methods and existing learning activities around cavity preparation, particularly for undergraduate Year 1 BDS students [32]. In the past, clinical skills teaching sessions using mannequin heads were the only simulators used in the curriculum (Phantom head). After validating the design of the haptic simulator [32,33], the training of artificial caries cutting skills was modified to study the effect of integrating haptics on students’ skills. The research procedures were carefully designed to ensure that students were not disadvantaged by the haptic interventions. Two studies were conducted with different cohorts of students. Additional teaching sessions were given after each study to ensure that learning outcomes required to successfully remove artificial caries were met.

### 2.1. The Participants, Settings, and the Artificial Caries Removal Task

All participants, including three tutors, consented to participate in this study (University research ethics approval reference CREC 06/07-222). The students’ average grades in relevant subjects were A for both groups. The students average age was 18. Two hundred and sixty-four (264) out of a total of two hundred and seventy-seven (277) enrolled students participated from two Year 1 cohorts. The students in each study were randomly assigned to the phantom and haptics condition groups. Table 1 shows that 138 (Study 1) and 126 (Study 2) students participated. The students had had no experience of using a drill nor had they held a dental handpiece before the study commenced. A formal sample size calculation was not performed as the sample size was limited by two factors: the size of the student cohort and the availability of the haptic devices.

One hundred and thirty-eight (138) (Study 1) and one hundred and twenty-six (126) (Study 2) students participated in the research. There were only enough devices for one third of each cohort of students to be assigned to the experimental group. The control group (phantom condition) completed cavity preparation tasks using the traditional phantom head facilities and plastic teeth; and the experimental group (haptics condition) operated the computer-based and haptics components on virtual teeth (see more details in the next section). With a limited number of haptic dental task trainer simulators available, two-thirds of each cohort were randomly assigned to the phantom head and one-third to the haptics condition. The students attended alternate Wednesday sessions and worked in pairs in either the clinical skills lab or the haptics lab (depending on their assigned group).

Two types of simulators were utilized in this study: (1) phantom-head simulator and (2) haptic dental task trainer simulator (haptics). Their designated tasks were equivalent (see Figure 1, inset pictures of the tasks). This section discusses their design features and corresponding operations.

The ADEC dental chair simulator (phantom-head), used in this study, has a mannequin head with rubber cheeks, replaceable plastic jaws and replaceable plastic teeth. This simulator also uses actual dental instruments including drills. In using this simulator for the purpose of the study, the students were only given two types of burs to use for slow and fast handpiece. The students used Frasaco plastic teeth designed for the purpose of the study, which have artificial plastic occlusal caries of varying complexities. The students sat upright in the desired position wearing protective glasses and operated by looking down into a mannequin head. The students could move their head to see inside the mouth but could also move around the mannequin head.

The haptic virtual reality dental task trainer simulator provides multimodal feedback particularly touch feedback (haptics) via an actual handpiece, which has an electronic component attached at one end that tracks the angulation and movement the handpiece makes and the other end has a steel ball that can be magnetically attached to the haptic device. There is also a dental mirror handle with similar attachments and function described above. Both these physical instruments are graphically modelled and are displayed on the 3D monitor together with the graphics of the selections of tasks showing teeth and mouth, as well as the drill and the sound it makes. As a user operates on realistic images of teeth and mouth, they change in orientation in respect to the natural movement of the user’s head. The students wear 3D glasses which are tracked by a camera to enable the mouth image to be viewed at different angles relative to the student’s head position. Additionally, to have an effect of co-location, i.e., perceiving that their hand is exactly in the same position as where they are operating in the 3D space, the monitor was configured with a mirror so that the user operates the system with the preferred ergonomics, as during the operation when using the phantom head. The only difference is that the mirror constrains the user from moving forward and bending their back to move closer to the ‘patient’s head’. The student can reposition the 3D head as if they moved their position with respect to the ‘patient’s head’.

### 2.2. The Learning Tasks and the Assessment of Removal of Artificial Caries

There were three teaching sessions spread over two months. During each session, the participants in both conditions were given tasks involving the removal of artificial carious lesions on an artificial tooth. The artificial teeth, both plastic and haptic, have three layers representing the enamel, dentine and pulp (see Figure 1, for example, for virtual and plastic teeth with artificial caries).

Session 1 involved working on a practice tooth which had no caries, and Task 1 was to remove the artificial caries, which is occlusal circular in shape of no more than 0.5 mm just into dentine. This session also involved familiarization with the corresponding technology. Sessions 2 and 3, respectively, involved working on a lower-left tooth located in a jaw with the size of caries ranged from 3 to 4 mm occlusal area and 2 to 3.5 mm depth.

After the three sessions, all the students were assessed using a plastic test tooth with a size and depth within the range of the difficult Task Teeth 3 (see San Diego and colleagues [32] for more details about the tasks).

A student’s performance is rated on five criteria

A.Choosing the angle of entry to the mouth used in gaining access to the artificial caries on a lower left first molar (tooth 36) in a jaw.B.Holding the instruments appropriately.C.Removing of artificial caries from the cavity wall.D.Removing of artificial caries cavity floor.E.Avoiding pulp exposure.

In scoring criteria, A and B, two assessors used the ‘glance and grade’ assessment technique using the scoring rubric provided. Criteria C, D and E were scored using the clinical operations performance Criteria A cavity preparation performance scoring rubric, based on Jokstad and Mjör [34], which was developed and validated. The details of the scoring rubric are provided in the following section.

### 2.3. Post-Test Only Cluster Randomized Controlled Design and Statistical Analysis

A post-test only data collection was conducted and had a cluster randomized controlled design. The performance of the two groups in removing the artificial caries from the test tooth was scored from 1 as a low score to 3 as a high score. Each score has a qualitative descriptor; take for example, scoring criterion D, on how well the student remove artificial caries on the cavity floor: 

Score of 1 = approximately two mm area of more artificial caries left.Score of 2 = approximately less than two mm area of more artificial caries left.Score of 3 = no artificial caries left. For more details of the scoring rubric, see San Diego and colleagues [32].

The total score that can be received by each student for each criterion ranges from 5 to 15. The students were given 15 min to complete the test. During the test session, as the students perform the task, four tutors scored the students on criteria A and B. The students took the test in groups of 24 so the tutors can rate their performance within the first few minutes of the session. The tutors were not told which simulator condition group the students belong to. Criteria C to E are rated after the session by a staff member. This staff member’s rating was calibrated with those of the tutors (see San Diego and colleagues) [32].

In the two studies conducted, the only difference is that the Task Tooth 3 and the test tooth in Study 1 were swapped in Study 2. The tutors believe that the Task Tooth 3 in Study 1 is slightly more difficult than the test tooth as the caries is closer to the pulp. The authors taught that it would be interesting to see whether there was a difference in the results upon replicating the study.

### 2.4. Statistical Analysis

Scores obtained by students were analysed between conditions in accordance with the following:The total number of students obtaining a total score of 5 to 15.The total number of students obtaining a rating of 1, 2 or 3 on each of the criterion.

These scores were analysed using Chi square and Fisher’s exact test as appropriate to the nature of the data.

During the sessions, the students were also given a session worksheet which contains instructions for the task and questions around the operations of the technology they are using and on cavity preparation. The participants were asked: (For the phantom head condition) to explain in detail how practicing with a plastic tooth helped or hindered them in completing the task.(For the haptics condition) To explain in detail how practicing with a haptics dental task trainer simulator helped or hindered them in completing the task.

These qualitative data were transcribed and used to explain the quantitative findings of this study.

## 3. Results

The results of Study 1, followed by Study 2, are presented below and are discussed using the qualitative data from the responses provided in the worksheets. The results are structured and tabled to allow the comparison of the frequency of scores between each criterion and/or between the phantom head condition (control group) and the haptics condition.

### 3.1. Study 1 Results

In comparing the performance between the two conditions, the difference is not significant (chi^2^ = 5.71 with *p* = 0.222) when examining the frequency of total scores obtained by participants (shown in Table 2A). However, although not statistically significant, the total number of participants who obtained scores of 14 and 15 is noticeably higher for the phantom head condition compared to the haptics condition. In grouping the frequency of total scores obtained from 11 to 13 and 14 to 15 between simulator conditions (as shown in Table 2B), a statistical difference using Fisher’s exact test (*p* = 0.03) can be seen between the haptics condition and phantom head condition in the frequency of a total score of 13 or below gained vs. a total score of 14 and above gained. The number of participants obtaining a total score of 13 or below is statistically greater for the haptics condition (52%) compared to 34% for the phantom head condition; whereas the number of participants obtaining a score of 14 or above is statistically greater for the phantom head condition (66%) compared to 48% for the haptics condition.

The difference in performance between the two groups may have been attributed to a specific criterion. In comparing the performance on individual criteria (Table 3A), no statistical significance in the frequency of students receiving a rating of “1”, “2” or “3” (all *p* values are greater than 0.05) between the two groups. However, it is noticeable that there are only a few individuals who received a rating of “1” between the two groups. So, ratings of “1” and “2” were combined and were recoded as “1” and rating of “3” was recoded as 2 (Table 3B). For each criterion between simulator groups, a significant difference (*p* = 0.018) was evident in Criterion B. In the phantom head condition, more students attained a rating of “2” compared to the haptics in holding the instruments appropriately, as rated by the two teachers observing students whilst performing cavity preparation during the test.

### 3.2. Study 2 Results

As described in Section 2.3 there was a small change in the procedure for Study 2. The tutors perceived that the test tooth in this study (the session 3 tooth task) was slightly more difficult than in Study 1 as the caries is closer to the pulp. The study was conducted to further validate the Study 1 results given a slightly more difficult plastic test tooth.

The total scores obtained by participants were from 10 to 15. Surprisingly, in comparing the total scores obtained by participants performance between the two groups, the difference is significant (chi^2^ = 13.42 with *p* = 0.02 as shown in Table 4A. This is the same when scores from 10 to 13 were compared with those with scores of 14 and 15 (as shown in Table 4B). The number of participants obtaining higher scores is statistically greater for the phantom head condition compared to the haptics condition.

Again, to check the performance between the two conditions in each criterion, in comparing the participants with a rating of “1” and “2” and “3” for each criterion (A to E) in the haptics and phantom conditions (see Table 5A), a strong statistical difference (*p* = 0.024) can be seen for criterion B (supporting Table 3B). The same statistically significant difference was found when the ratings were recoded to “1” (combining those with ratings “1” and “2”) and “2” (those with rating “3” in Table 5B). In Criterion B, similar to Study 1, more students in the phantom condition attained a rating of “2” compared to the haptics condition in holding the instruments appropriately, as rated by the two teachers observing students whilst performing cavity preparation during the test.

### 3.3. Analysis of the Worksheets

To explain the similarities and differences in performance the participants’ feedback in the worksheets were considered.

Typical feedback provided by phantom participants on how practicing with a plastic tooth helped or hindered them in completing the task. Most participants agreed positively about the use of phantom head in performing the task.

“I feel the plastic tooth was very representative of a real cavity preparation, as it was very life-like and three-dimensional, so it enabled a realistic experience. I think it’s very useful to practice handling the drill without making errors on real patients.”(Student 3)

“Practicing with a plastic tooth has helped me in completing the test. It gave me a feel of how the drill cuts through the material and at which angle to drill.”(Student 108)

Typical answers provided by haptics participants feedback on how practicing with a plastic tooth helped or hindered them in completing the task. The answers below show mixed opinions. However, the frequency of positive opinions outweighs the negative. Most students felt that although the haptic dental task trainer simulator is harder to use than the phantom head, it subsequently made the handpiece of the latter easier for them to control.

“It may have helped as it feels a lot heavier and less accurate to use, so when given an actual drill this felt lighter, more ‘flexible’ and easier to use.”(Student 93)

“Haptics was a lot harder to drill out the cavity, I think this made the plastic tooth cavity prep easier by being more precise in haptics.”(Student 32)

“Helped as an indicator of what to expect with drilling. I liked the zooming into a tooth and analyzing your work as you went along. It was also a lot harder, which put you in better (stead) for real-life drilling.”(Student 21)

“It did not help as there was greater (pressure) to your hand moving the bur, so you had to push down harder, this did not transfer to the bur and as a result I hit pulp. Also, the set up in the haptics room is more computer-based so the new environment was difficult...”(Student 11)

“Hindered me because I did not know how a real bur works and how fast/slow it drills through the carious lesion.”(Student 24)

## 4. Discussion

This study examines whether using a haptic dental task trainer simulator is equally as beneficial for dental students as the established traditional phantom head simulator: The analysis of results obtained in Study 1 and 2 indicates that the overall learning of cavity preparation was evident in both the haptics and phantom head conditions. This finding is similar to what others found about the use of haptics in another clinical context [26,27]. Our research corroborates Dwisaptarini, A., and colleagues [26], in that both simulators had equivalent effects, in particular with our findings, regarding the removal of artificial carious lesion. Moreover, Vincent and colleagues cross-sectional study evaluating VirTeaSy haptic simulator indicated similar benefits on learning curves to that of traditional training methods [27]. In teaching pre-clinical coronal cavity preparations, the use of the Simodont virtual simulation system was found to improve dental students’ scores when combined with the traditional phantom head simulator [35]. This suggests that haptics have potential as an alternative simulator or at least adjunct with traditional mannequin simulators to provide effective pre-clinical training in dental education. Although, in Study 2, it seemed that students may perform better in removing artificial caries with the more challenging task when using phantom simulators. However, this may be due to the low level of realism in the plastic teeth where the pulp is located very close to the site of the ‘lesion’. The qualitative data suggests that students find the haptic experience in terms of the ‘feel’ in drilling is different compared to using an actual drill with a plastic tooth. Both groups seemed to believe that the phantom simulator felt more ‘real’ than the haptic drilling. Recreating a clinical experience in a simulated environment is complex and difficult, which should be taken into account, and the benefits it provides to clinical training is very important to explore. The qualitative comments also seemed to suggest that the haptic operation of a dental handpiece is more difficult compared with the operation of an actual dental handpiece in the phantom head. However, students in the haptics condition group seemed to perceive this difficulty as beneficial as they move from using haptics to the phantom head. This requires further validation and study. The notion that students perceive some aspects of the conventional training simulators as more realistic is not uncommon; Leung et al., 2021, found that 79% of dental students agreed that traditional simulations ‘felt more realistic’ compared to the VR haptic-based simulator Simodont. Nevertheless, generally there is a positive regard for haptics by dental students, which suggests this technology has a firm place alongside traditional simulators [19,36].

As illustrated in Table 3B and Table 5B, a higher frequency of students in the haptics condition received a lower score with regards to criterion B (holding the instruments appropriately), compared to those in the phantom head group (control group). It is an incidental finding that there is some suggestion that the haptics group were less likely to hold their instruments correctly compared with the phantom group. This requires further study. However, extra care needs to be considered by the researcher who pursues this. The tactile fidelity of haptic devices may differ, which means that it can be difficult to generalize and attribute the effect in handling instruments to haptic feedback. The ‘glance and grade’ may also be a contributing factor. Not holding the instruments correctly could be partially attributed to the participants’ visual-spatial awareness, as discussed earlier [30]. A high number of students in the haptics condition obtaining lower scores may possess restricted abilities to accurately process the positioning of the haptic device in concordance with the simulation. This inability can result in the incorrect handling of the haptic device navigating the simulated instrument.

## 5. Conclusions

Equal benefit can be seen when students are trained by either haptic or traditional simulations in acquiring skills and understanding concepts relevant to the removal of artificial carious lesions. However, there were small differences in ratings of performance for a more demanding task. Students in the haptic group were also less likely to be perceived to be ‘holding the instrument appropriately’. These findings require further investigation to understand whether the difference is relevant to actual clinical practice, as in this study the plastic caries, dentine and pulp, and the exposure of pulp with regard to tolerable thickness of dentine left may not be the same in real patient tissue. Moreover, a limitation in this study is that one type of plastic test tooth located in a single position (lower left first molar (tooth 36)) was used. Student scores in all the categories examined may differ if trained on different sized and shaped plastic teeth in varying positions in the mouth. Although over 10 years has passed since the validation of the design of the VR haptic simulator was completed, the results obtained in this study clearly shows it is still highly relevant and applicable in today’s research.

## Figures and Tables

**Figure 1 dentistry-10-00198-f001:**
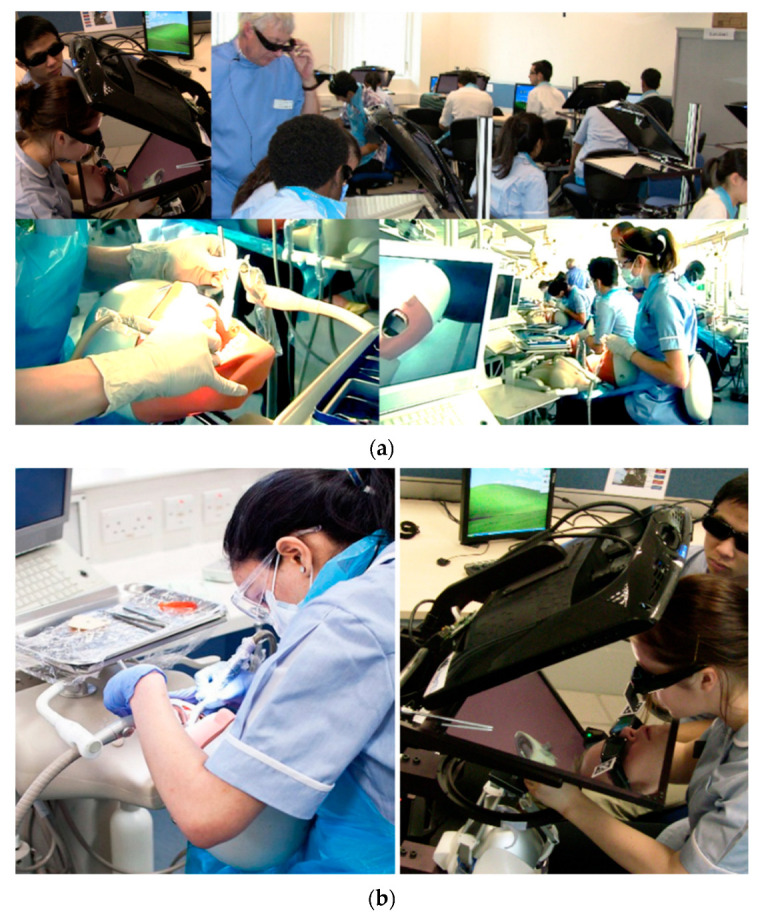
(**a**) The phantom and the haptics Clinical Skills Laboratories. The top figure shows the teacher glancing over the students working on an example haptic tooth task. The bottom figure shows students performing the removal of artificial caries on a plastic tooth. (**b**) Students training on the phantom head simulator (**left**) and VR haptic simulator (**right**).

**Table 1 dentistry-10-00198-t001:** The number of participants in the Phantom and Haptic Conditions for Study 1 and 2.

Condition	Study 1 (Total = 138)	Study 2 (Total = 126)
Phantom	*n* = 94	*n* = 79
Haptics	*n* = 44	*n* = 47

**Table 2 dentistry-10-00198-t002:** A: Comparison on overall performance of cavity preparation (scale range 5 to 15) (*n* = 138, Study 1). B: Re-coded comparison on overall performance of cavity preparation (scale range 5 to 15) (*n* = 138, Study 1).

A				
Scores	Haptics (*n* = 44)		Phantom (*n* = 94)	
11	1	2%	2	2%
12	3	7%	7	7%
13	19	43%	23	24%
14	14	32%	47	50%
15	7	16%	15	16%
Mean	13.52		13.7	
chi^2^ = 5.71 (*p* = 0.222)				
**B**				
**Score**	**Haptic**		**Phantom**	
11 to 13	23 (52%)		32 (34%)	
14 to 15	21 (48%)		62 (66%)	
Fisher’s Exact Test (*p* = 0.03)				

**Table 3 dentistry-10-00198-t003:** A: Comparison of performance on individual criteria (*n* = 139, Study 1). B: Re-coded comparison of performance on individual criteria (*n* = 139, Study 1).

A								
	Haptic Condition (*n* = 44)			Phantom Condition (*n* = 94)				
	1	2	3	1	2	3	chi^2^	*p*
A	1 (2%)	30 (68%)	13 (30%)	4 (4%)	62 (66%)	28 (30%)	0.35	0.84
B	2 (4%)	21 (48%)	21 (48%)	2 (2%)	28 (30%)	64 (68%)	5.34	0.69
C		2 (4%)	42 (96%)		11 (12%)	83 (88%)		0.11 ^a^
D		4 (9%)	40 (91%)		5 (5%)	89 (95%)		0.194 ^a^
E	1 (2%)	1 (2%)	42 (96%)	1 (1%)	2 (2%)	91 (97%)		0.86 ^a^
**B**								
	**Haptic Condition (*n* = 44)**			**Phantom Condition (*n* = 94)**				
	**1**		**2**	**1**		**2**	** *p* **	
A	31 (70%)		13 (30%)	66 (70%)		28 (30%)	0.572	
B	23 (52%)		21 (48%)	30 (32%)		64 (68%)	0.018	
C	2 (4%)		42 (96%)	11 (12%)		83 (88%)	0.11	
D	4 (9%)		40 (91%)	5 (5%)		89 (95%)	0.194	
E	2 (4%)		42 (96%)	3 (3%)		91 (97%)	0.327	

^a^ Fisher Exact Test.

**Table 4 dentistry-10-00198-t004:** A: Comparison on overall performance of cavity preparation (scale range 5 to 15) (*n* = 138, Study 1). B: Re-coded comparison on overall performance of cavity preparation (scale range 5 to 15) (*n* = 138, Study 1).

A		
Scores	Haptics (*n* = 47)	Phantom (*n* = 79)
10	3 (6%)	1 (1%)
11	4 (9%)	4 (5%)
12	9 (19%)	8 (10%)
13	19 (40%)	21 (27%)
14	8 (17%)	35 (44%)
15	4 (9%)	10 (13%)
Mean	13.46	12.79
	chi^2^ = 13.42	(*p* = 0.02)
**B**		
**Score**	**Haptic**	**Phantom**
<=13	35 (74%)	34 (43%)
>=14	12 (26%)	45 (57%)
Fisher Exact Test (*p* = 0.0)		

**Table 5 dentistry-10-00198-t005:** A: Comparison of performance on individual criteria (*n* = 126, Study 2). B: Re-coded comparison of performance on individual criteria (*n* = 126, Study 2).

A								
	Haptic Condition (*n* = 47)			Phantom Condition (*n* = 79)				
	1	2	3	1	2	3	chi^2^	*p*
A		31 (66.0%)	16 (34%)	3 (4%)	37 (47%)	39 (49%)	5.367	0.068
B	7 (15%)	30 (64%)	10 (21%)	2 (3%)	52 (65%)	25 (32%)	7.463	0.024
C	5 (11%)	7 (15%)	35 (74%)	5 (6%)	6 (8%)	68 (86%)	2.697	0.26
D	3 (2%)	2 (4%)	42 (94%)	2 (3%)	1 (1%)	76 (96%)	2.355	0.308
E	1 (2%)	2 (2%)	44 (96%)			79 (100%)	5.166	0.076
**B**								
	**Haptic Condition (*n* = 47)**				**Phantom Condition (*n* = 79)**			
	**1**		**2**		**1**		**2**	
A	31 (66%)		16 (34%)		40 (51%)		39 (49%)	
B	37 (79%)		10 (21%)		54 (68%)		25 (32%)	
C	12 (26%)		35 (74%)		11 (14%)		68 (86%)	
D	5 (6%)		42 (94%)		5 (5%)		89 (95%)	
E	3 (4%)		44 (96%)				79 (100%)	

## Data Availability

Not applicable.

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
