# Peer review of "Learning Clinical Skills Using Haptic vs. Phantom Head Dental Chair Simulators in Removal of Artificial Caries: Cluster-Randomized Trials with Two Cohorts’ Cavity Preparation"

_dentistry, 2022, doi:10.3390/dj10110198_

Round 1

Reviewer 1 Report

Dear Authors,

After reading your article very carefully I think it is an interesting article in the educational area. The experiment is well designed and the statistical approach is correct. However, the manuscript could be deeply improved if you paid more attention to the redaction style and reorganized part of the introduction, discussion and conclusions. There are some little mistakes that should be solved before taking the manuscript into consideration for final publication. I have attached a pdf document of your draft with my specific considerations in it

Reviewer 2 Report

This paper presents the comparison results of learning clinical skills of dental course students using a haptic/VR dental simulator vs a mannequin-head/physical dental chair simulator in removal of artificial caries, i.e., cavity preparation. Students were classified into two cohorts. Across both cohorts, there was no difference in the quality of cavity cut, but, students’ technique differed. No difference was seen across both cohorts in the quality of cavity cut for a simple preparation, but, students in the haptic condition performed less well in the more demanding task. Moreover, students in the haptic group were also less likely to be perceived to be ‘holding the instrument  appropriately’. In this way, the paper includes these findings and suggests further investigation is needed into the differences in handling of instruments and level of clinical task difficulty between the simulators.

The paper seems written well. I have just  a request to improve the paper. It is difficult to understand the difference between a haptic/VR dental simulator and a mannequin-head/physical dental chair simulator. So, it is better to show the pictures/figures of the two simulators in 2.2 besides Figure 1. 
